# Design of a Compact Ultra-Wideband Microstrip Bandpass Filter

Chen Li [1], Zhong-Hua Ma [1,2,*], Jia-Xiang Chen [1], Meng-Nan Wang [1] and Jian-Mei Huang [1]

1 School of Marine Information Engineering, Jimei University, Xiamen 361021, China; lichen0701@jmu.edu.cn (C.L.); chenjx7335@jmu.edu.cn (J.-X.C.); 202211810012@jmu.edu.cn (M.-N.W.); 202021112035@jmu.edu.cn (J.-M.H.)
2 Fujian Provincial Key Laboratory of Oceanic Information Perception and Intelligent Processing, Xiamen 361021, China
* Correspondence: mzhxm@jmu.edu.cn

**Abstract:** A compact ultra-wideband bandpass filter based on a multilayer printed circuit board (MPCB) structure is proposed in this paper. RO4450F prepreg is used to bond three RO4350B dielectric substrates with different thicknesses in the MPCB structure. The upper surfaces of the three dielectric substrates are respectively provided with copper-coated structures with different patterns. The blind holes and the defected ground structure (DGS) are added to the MPCB of an ultra-wideband bandpass filter. Two groups of loaded quarter-wavelength terminal-open stubs introduce two transmission zeros, which improves the roll-off rates and stopband characteristics, while simple DGS composed of rectangular slots introduces two resonance points in the passband to improve the return loss. Simulation and measurement are consistent. The insertion loss at the center frequency of 12.795 GHz is 0.58 dB and the fractional bandwidth of 3 dB is 40.33% from 10.215 GHz to 15.375 GHz. This bandpass filter can be widely used in wireless and satellite communication.

**Keywords:** defected ground structure; insertion loss; return loss; transmission zero; ultra-wideband bandpass filter

## 1. Introduction

As wireless and satellite communication technologies, radio frequency circuits and communication systems have become increasingly complex, the whole circuits tend to be integrated and miniaturized. The interference between communication systems is becoming more and more serious, while the ultra-wideband (UWB) wireless system has the advantages of high-speed data transmission, strong anti-interference performance and low transmission power. UWB wireless communication has attracted extensive attention, and more and more ultra-wideband filters have been proposed. One of the methods to design a UWB bandpass filter is to use multimode resonators (MMR), and even–odd mode analysis is often used to analyze the transmission characteristics of such filters. In Refs. [1–3], scholars adopted the design method of multimode resonators and used the characteristics of odd and even resonance modes to determine the center frequency and adjust the passband bandwidth. Combined with different structural designs and material choices, wideband or ultra-wideband passband characteristics can be achieved. An equivalent circuit is an effective method to help analyze the transmission characteristics and resonance modes of an ultra-wideband bandpass filter when designing it. In Refs. [4,5], the transmission function and frequency characteristics are observed by circuit simulation, and the position changes of the center frequency and transmission zeros are predicted by adjusting the equivalent capacitance and inductance of the microstrip line length. This method has also been reported in many studies. In Ref. [6], the equivalent circuit model is extracted from the topology of the spatial filter under the conditions of tight coupling and loose coupling, which are used to better understand the filtering mechanism and

predict the electromagnetic behavior of the filter. Then the circuit model can be further optimized through simulation and calculation. In Ref. [7], an ultra-wideband bandpass filter is designed based on the coupled branch lines, and the even and odd resonance modes of the filter are analyzed by using the equivalent circuit to suppress the stopband harmonics. At the same time, a wide reconfigurable bandwidth range is realized by variable capacitors. Ref. [8] reported a reconfigurable UWB bandpass filter, which was a cascaded design consisting of a reconfigurable lowpass filter and a reconfigurable highpass filter and had excellent out-of-band suppression. Because of the cascaded structure itself, the ultra-wideband filter designed by this method was relatively large in size, complex in structure and large in insertion loss (IL). In Ref. [9], a bandpass filter with a 3 dB fractional bandwidth (FBW) of 32% was designed by using two quarter-wavelength resonators in a comb-line structure. In Ref. [10], a UWB filter based on a quarter-wavelength stepped impedance resonator was reported. The 3 dB fractional bandwidth is 55%, but the insertion loss is also large.

Many new technologies and materials have been developed for the miniaturization design of ultra-wideband filters. Qian Kewei et al. proposed an ultra-wideband filter with three-layer structure manufactured by low-temperature co-fired ceramic (LTCC) technology with a fractional bandwidth of 85.2% at the center frequency of 3.2 GHz, and the size of the whole filter was only $5 \times 4.5 \times 0.65$ mm$^3$ [11]. An ultra-wideband bandpass filter with a 3 dB fractional bandwidth of 75% with a center frequency of 1.8 GHz was reported in [12]. Thanks to the convenience of low-temperature co-fired ceramic technology, this filter with an eight-layer structure design was only $5.2 \times 3.6 \times 1.2$ mm$^3$ in size. By introducing an appropriate coupling structure, a steep transition band was realized, but the in-band reflection was relatively large. In Ref. [13], liquid crystal polymer (LCP) technology was used to realize the integration of wideband coupling patch, embedded quarter-wave resonators, bandpass periodic units and defected ground structure (DGS), and a design of an ultra-wideband bandpass filter with high selectivity was completed. In Ref. [14], a UWB filter with 180% fractional bandwidth at the center frequency of 10.5 GHz, which was integrated by multilayer liquid crystal polymer technology, was reported. Liquid crystal is mainly used in the design of reconfigurable filters, and its manufacturing process is complicated.

To realize the miniaturization of the filter, a novel multimode double-ring ultra-wideband bandpass filter based on high-temperature superconducting material was proposed [15]. The filter, with a size of $21.95 \times 16.46$ mm$^2$, was realized on a MgO substrate with a thickness of 0.5 mm. Its 3 dB FBW reached 125.3% and the minimum insertion loss in the passband was less than 0.42 dB. Filters made of high-temperature superconducting material film have small insertion loss, steep transition band and good out-of-band suppression characteristics, but the manufacturing process is complicated. In Ref. [16], a miniaturized on-chip bandpass filter based on GaAs was presented. By introducing coupling capacitor and series resonance circuits, transmission zeros were generated, and out-of-band rejection was improved. Its size was $0.8 \times 0.5$ mm$^2$, and the relative bandwidth was 39.02%. In Ref. [17], a method of preparing lithium niobate single-crystal film on multilayer oxide film was proposed to obtain a bulk acoustic wave filter with oxide Bragg reflector. The filter had a broadband bandwidth of 10.3% at 3.128 GHz, but its size was reduced to only $0.6 \times 0.4$ mm$^2$, which was expected to be applied in 5G systems. In addition to the new materials mentioned above, graphene is also a material for designing novel filters. By adjusting the structure of graphene layer, its chemical potential and other characteristics are changed, so that the center frequency of filters and antennas is reconfigurable [18]. At present, existing graphene-based filters and antennas have been designed and applied in conventional ultra-wideband and terahertz frequency bands [19], and they also have broad design and application prospects in the future.

The technology of multilayer printed circuit board (MPCB) can also be used to realize compact multilayer microwave circuits, which is a low-cost method based on dielectric materials with relatively low dielectric constant and good via support. Compared with

LTCC, LCP and other new material technologies, devices made of PCB can be much more easily integrated with other electronic devices and packaged into existing microwave systems in chips. At the same time, the process of multilayer PCB can also enable filter design of multilayer structures in three-dimensional space, which meets the requirements of miniaturization and structural diversification of new filters.

Some works in the literature reported the design of UWB bandpass filters based on MPCB. In [20], a dual-band bandpass filter with two transmission paths was designed by using substrate integrated coaxial lines and a reusable semi-open transmission line. The filter adopted a design of vias and blind holes on three layers of RO4350 substrate and achieved a relative bandwidth of 58%. In [21], Yang Ling et al. proposed a filter with complementary split-ring resonators that was based on the folded substrate integrated waveguide structure. The filter used two layers of Rogers5880 dielectric substrates (Chandler, AZ, USA), with a relative bandwidth of 35.5% and good stopband characteristics. In [22], a packaged bandpass filter based on MPCB was proposed by the Rao Yun-Bo team, which was suitable for 5G millimeter wave applications. The filter had a fractional bandwidth of 27.9% in the band of 28 GHz, and its out-of-band harmonic suppression level was higher than 24 dB up to 110 GHz.

In this paper, a novel UWB bandpass filter based on MPCB structure is proposed. The bandpass filter is made of Rogers RO4350 dielectric substrates and pieces of RO4450F prepreg. The upper surface of each dielectric substrate is coated with copper according to different design patterns and is grounded through two blind metalized holes. The transmission zeros and the resonance points are produced by using two groups of quarter-wavelength terminal-open stubs and DGS, respectively, which enhances the stopband suppression and reduces the return loss (RL) in passband of the filter. The measurement results show that the 3 dB FBW of the filter is 40.33%, the in-band reflection is better than 18 dB and the insertion loss at the center frequency of 12.795 GHz is 0.58 dB. Compared with some UWB bandpass filters designed with single-layer planar structures, the proposed multilayer laminated filter is relatively small in size. The three-layer coupling patterns of the filter in this paper are all simple SIR structures and extensions. The design of the DGS consists of three simple rectangles, all the design patterns are not complicated and there are only two blind holes in the interlayer via design. Generally speaking, the design method is relatively simple.

## 2. Filter Design

Figure 1 shows the laminated structure and design patterns of each layer, which consists of three layers of Rogers RO4350 dielectric substrate and four layers of RO4450F prepreg. Rogers RO4350 dielectric substrate has a relative dielectric constant of 3.66 and a loss tangent of 0.009. The thicknesses of the three dielectric substrates are 0.1016 mm, 0.254 mm and 0.508 mm respectively. Thermosetting resin bonding material is selected according to the processing requirements when processing MPCB structure. In this paper, RO4450F thermosetting resin bonding sheet compatible with R04350B is selected, with a thickness of 0.1016 mm, a dielectric constant of 3.52 and a loss tangent of 0.004. Two pieces of RO4450F need to be filled between every two substrates. The thickness of the copper skin of the top layer and the grounding layer is 0.035 mm, and the thickness of the copper skin of the middle two layers related to the lamination of RO4450F prepreg is set to 0.018 mm.

The volume of the filter is $20.8 \times 20 \times 1.34$ mm$^3$. The upper surfaces of three dielectric substrates are respectively coated with copper according to different design patterns, and four pieces of RO4450F with the same thickness are used for bonding them. Two blind holes are drilled in the wider part of the stepped impedance resonator (SIR) from the bottom ground plate through the third dielectric substrate to its upper surface. A DGS consisting of three rectangular slots is etched on the metal ground layer.

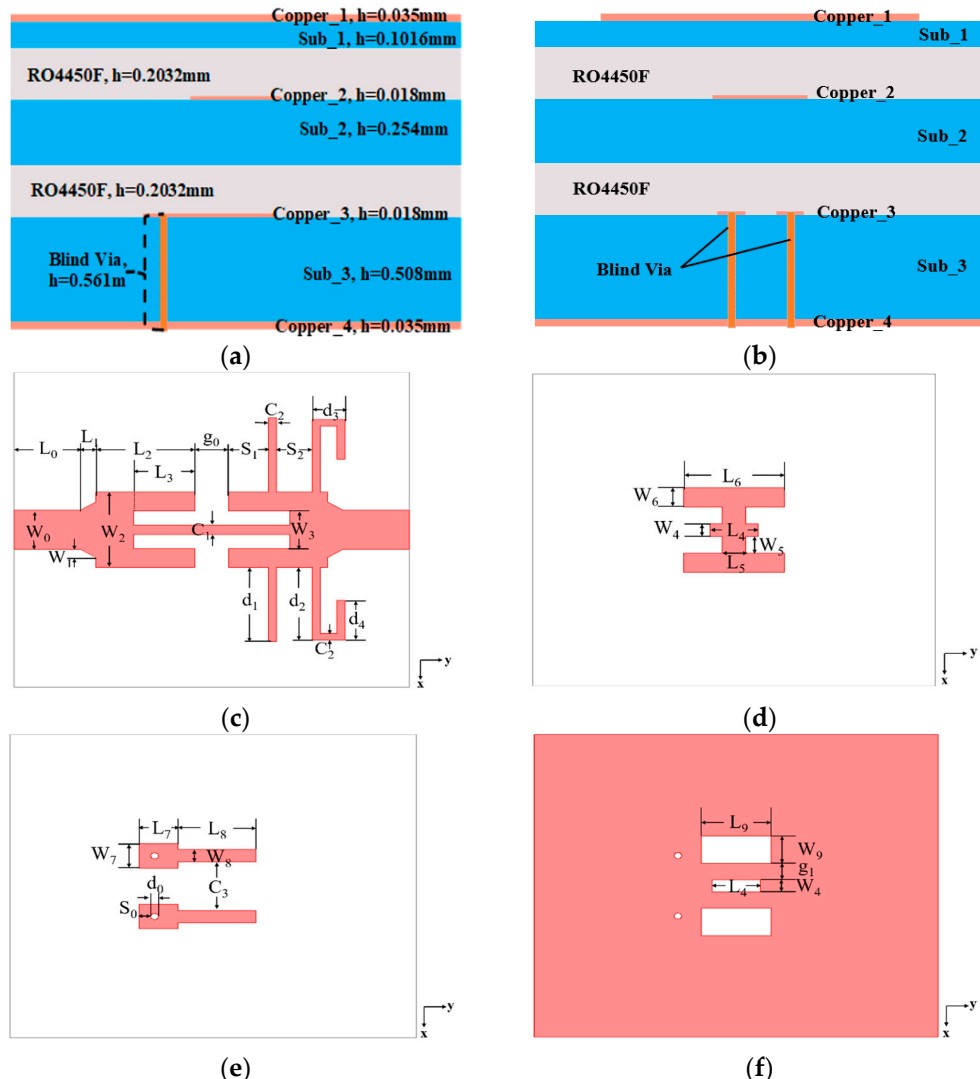

**Figure 1.** (**a**) Side view of filter structure; (**b**) Front view of filter structure; (**c**) Structure of top coupling patch, Copper_1; (**d**) Structure of second coupling patch, Copper_2; (**e**) SIR structure of the third layer, Copper_3; (**f**) Ground layer of DGS, Copper_4.

When processing, the multilayer PCB to be pressed filled with RO4450F is clamped first, and then the clamping is positioned. After that, the temperature-programmed operation is carried out. With the gradual increase of laminating temperature, the prepreg resin will flow with it, and it will be filled between the copper circuit patterns with the help of uniform pressure attached to the multilayer PCB to be pressed, so that there will be no gap between the substrates.

The whole design process includes four steps of optimization and improvement. Besides some marked design parameters being adjusted according to the actual processing conditions, the main optimization direction is to change the design structure of the top coupling patch and introduce the DGS composed of rectangular slots. Figure 2 shows the optimization process of the overall design of the filter. The curves of their corresponding transmission characteristics and reflection characteristics are shown in Figure 3.

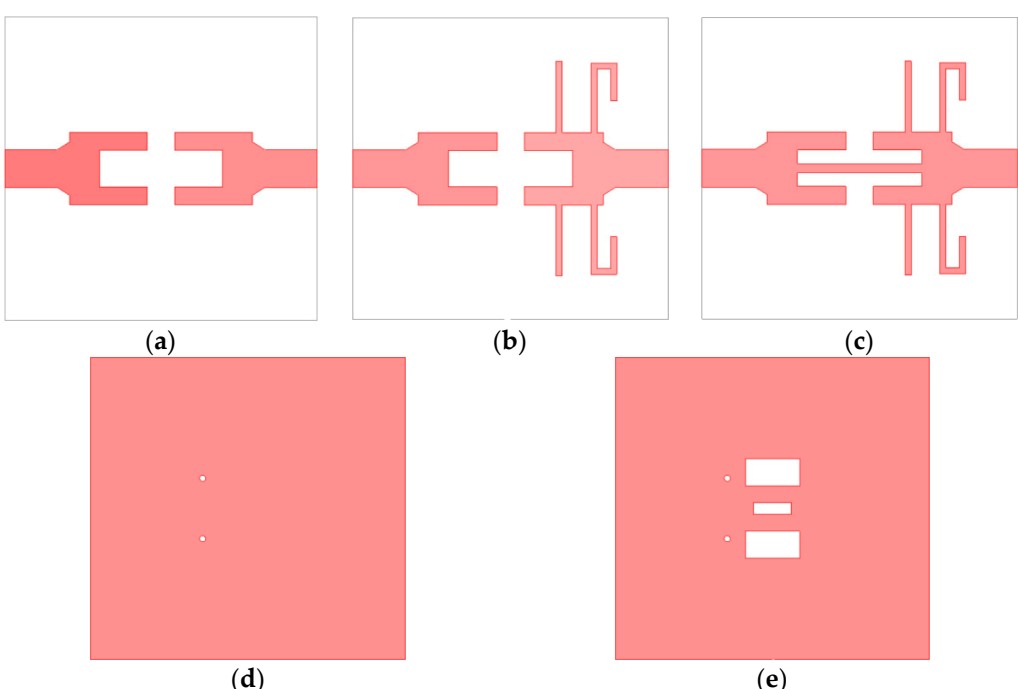

**Figure 2.** (**a**) Coupling pattern of two U-shaped microstrip lines; (**b**) Loading two groups of terminal-open stubs; (**c**) Adding central connecting microstrip line; (**d**) Ground plane with two blind holes; (**e**) Ground layer with defected structure.

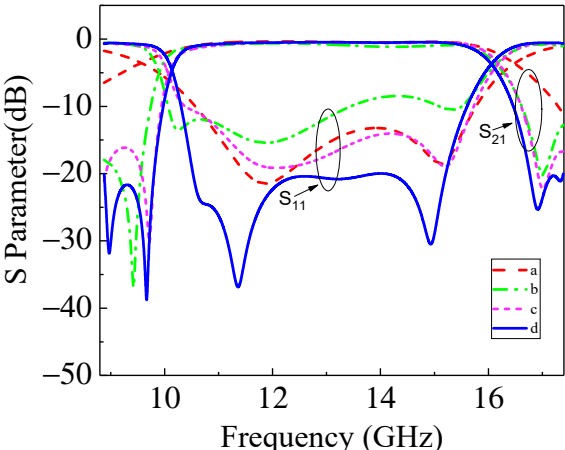

**Figure 3.** Comparison diagram of simulated reflection and transmission characteristic curves of structural evolution from Figure 2, in which curves (**a**) correspond to the structure of Figure 2a of two U-shaped microstrip lines; curves (**b**) correspond to the structure of Figure 2b of loading two groups of open stubs with different lengths on the top layer; curves (**c**) correspond to the structure of Figure 2c of adding central connecting microstrip; curves (**d**) correspond to the structure of Figure 2e of the ground layer with DGS.

Figure 2a shows the initial design of the top coupling patch. Its reflection characteristics and transmission characteristics correspond to the curves (a) in Figure 3. It can be seen that the roll-off rate on both sides of the filter passband is low at this time, and the stopband suppression is not good. To improve the filter's roll-off rate and optimize its stopband performance, two pairs of terminal-open microstrip stubs with different lengths are loaded on the U-shaped microstrip line at the output, as shown in Figure 2b. The corresponding curves of reflection coefficient and transmission coefficient are shown in curves (b) in

Figure 3. Two pairs of terminal-open microstrip stubs are used to enhance out-of-band attenuation and roll-off factor from the perspective of impedance analysis.

The transmission line input impedance is defined as:

$$Z_{in} = Z_0 \frac{Z_L + jZ_0 \tan(\beta l)}{Z_0 + jZ_L \tan(\beta l)} \tag{1}$$

where $Z_0$ is the transmission line of characteristic impedance. When the transmission line terminal is open-circuit, $Z_L \to \infty$.

By substituting $Z_L$ into (1), $Z_{in}$ is expressed as follows:

$$Z_{in} = -jZ_0 ctg(\beta l) \tag{2}$$

Here, $\beta = 2\pi/\lambda$. When the length of the stub line is a quarter-wavelength, the input impedance is the following:

$$Z_{in} = 0 \tag{3}$$

After a quarter-wavelength, the open circuit of the stub becomes a short circuit. Therefore, the quarter wavelength stub lines implement grounding.

Because the terminal-open stubs are quarter wavelengths, two transmission zeros outside the passband will be generated, which correspond to 9.4 GHz and 16.8 GHz. At the same time, considering the coupling between two quarter-wavelength terminal-open stubs, the two groups of stubs are set at an appropriate distance to prevent the interference of short-range coupling on the transmission signal. The curves (b) in Figure 3 show that the roll-off factor is greatly improved after loading stubs, and the stopband harmonic suppression performance is also optimized. Introduction of two transmission zeros optimizes the passband selectivity, but in-band reflection coefficient becomes slightly worse. Of course, the stopband rejection characteristics are also not good. In the subsequent design process, the stopband suppression characteristics would be optimized by adjusting parameters on the premise of maintaining the passband characteristics. Figure 4 is the current distribution of the top coupling patch after parameter adjustment. The positions of the transmission zeros have a slight frequency shift compared with the initial one. It can be seen that, at the frequency corresponding to the transmission zeros, two groups of open-circuit stubs have played their roles.

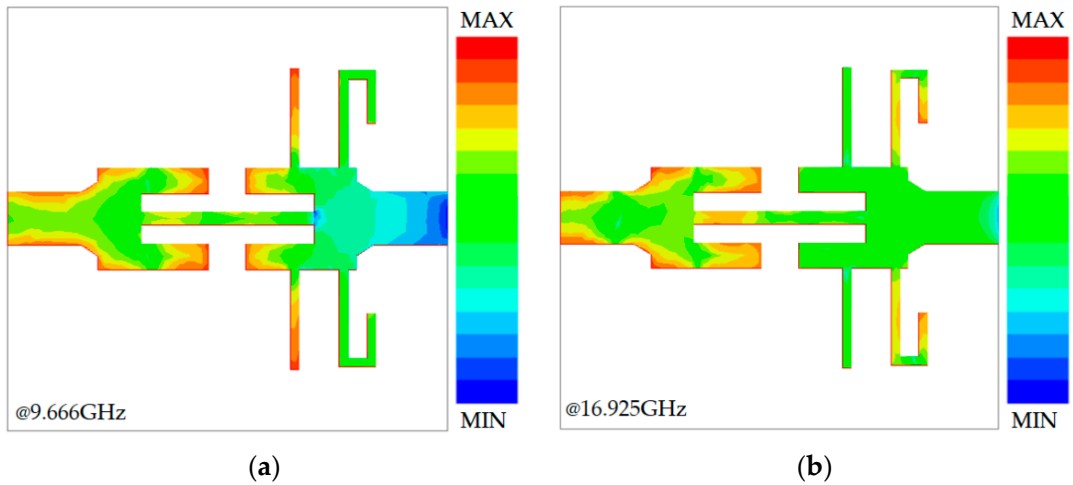

**Figure 4.** Current distribution of top coupling patch at the frequency of transmission zeros: (**a**) 9.666 GHz; (**b**) 16.925 GHz.

To optimize the reflection characteristics in the passband of the filter, a central connecting microstrip is added between two symmetrical U-shaped microstrip lines, as shown in Figure 2c, so that part of the energy of the input is not only coupled down through the

U-shaped microstrip line, but also directly transmitted to the U-shaped microstrip line on the other side through the central connecting microstrip. At the same time, part of the energy reflected from the lower layers will be transmitted through the central connecting microstrip, thus reducing the return loss of signal energy, corresponding to the curves (c) in Figure 3. We can see that in-band reflection has been obviously improved, the reflection coefficient of passband is optimized by 5dB to 8 dB, the roll-off coefficient of transmission characteristics increases, the low-frequency cut-off frequency of passband moves to a higher frequency and the bandwidth decreases slightly.

In order to further optimize the in-band reflection characteristics, a DGS composed of three rectangular slots is loaded on the bottom ground plate as shown in Figure 2e. DGS can improve current distribution and return loss, adjust cut-off frequency and resonance frequency and optimize stopband characteristics [23,24]. The curves of reflection and transmission characteristics after loading the DGS are shown in curves (d) in Figure 3. These three rectangular slots change the surface current path of the metal grounding layer. Two wider and longer rectangular slots on the outside introduce the low-frequency resonance point, while narrower and shorter rectangular slot introduces high-frequency resonance point. The reflection coefficient in the passband is optimized by changing the parameters to adjust the positions of the two resonance zeros. Finally, the reflection coefficients in passband are all less than $-20$ dB, and the two resonance points reach $-36.85$ dB and $-30.44$ dB at 11.361 GHz and 14.943 GHz, respectively. In order to obtain good flat passband performance, excellent out-of-band roll-off and harmonic suppression, a series of parameters of the proposed filter are optimized. By changing different structural parameters one by one, the influences of these parameters on the filter performance are studied. Finally, the optimal parameters of the filter in Table 1 are obtained.

**Table 1.** Optimal parameters of the proposed filter. (Unit: mm).

| $W_0$ | $W_1$ | $W_2$ | $W_3$ | $W_4$ | $W_5$ | $W_6$ | $W_7$ | $W_8$ | $W_9$ | $g_0$ |
|-------|-------|-------|-------|-------|-------|-------|-------|-------|-------|-------|
| 2.5 | 0.5 | 4.8 | 2.4 | 0.8 | 1.1 | 1.2 | 1.6 | 0.8 | 1.8 | 1.8 |
| $L_0$ | $L_1$ | $L_2$ | $L_3$ | $L_4$ | $L_5$ | $L_6$ | $L_7$ | $L_8$ | $L_9$ | $g_1$ |
| 3.5 | 0.8 | 5.2 | 3.2 | 2.5 | 1.2 | 5.2 | 2 | 4 | 3.6 | 1.1 |
| $d_0$ | $d_1$ | $d_2$ | $d_3$ | $d_4$ | $C_1$ | $C_2$ | $C_3$ | $S_1$ | $S_2$ | $S_3$ |
| 0.4 | 4.7 | 4.2 | 1.7 | 2.1 | 0.6 | 0.4 | 3.2 | 0.6 | 2.1 | 1.9 |

This bandpass filter can also be approximated using a Chebyshev function with a transfer function to analyze [25]:

$$|S_{21}(j\Omega)|^2 = \frac{1}{1 + \varepsilon^2 T_n^2(\Omega)} \tag{4}$$

where $\varepsilon$ is the ripple constant. Low cutoff frequency of the bandpass filter $f_{c1}$ = 10.215 GHz and high cutoff frequency $f_{c2} = (\pi/\theta_c - 1) \times f_{c1}$, where $\theta_c$ is the electrical length of the distributed parameter element. If the electrical length of the element is 71.7°, the high cutoff frequency is 15.375 GHz.

## 3. Parameter Analysis

Several key parameters are studied to further analyze the characteristics of this filter. The main parameters for analysis are the bottom width $W_2$ of the U-shaped microstrip lines on the top layer, length $L_6$ of the wider side of the second coupling patch, diameter $d_0$ of the blind holes on the third SIR branch lines, and width $W_9$ of the wider rectangular slots in DGS.

In Figure 5, the influence of length $L_6$ of the wider side of the second coupling patch on RL is shown. With the increase of $L_6$, the coupling area between the second coupling patch, the top coupling patch and the third-layer SIR branch lines gradually increases, thus improving reflection characteristics. When $L_6$ = 5.2 mm, it can be seen that the curve

of reflection coefficient between the two resonance points in the passband is less than −20 dB. It should be noted that the rectangular on the outside of the second coupling patch corresponds to the rectangular slots on the outside etched in the metal ground layer in the vertical position, the position of the resonance points in the passband will also be affected to some extent. $L_6$ = 5.2 mm is finally determined as the optimal value.

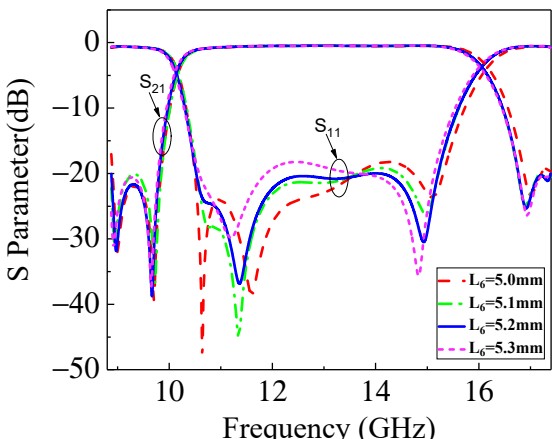

**Figure 5.** Simulation of S parameters when length $L_6$ of the wider side of the second coupling patch is varied from 5 mm to 5.3 mm, with a step of 0.1 mm increment.

In Figure 6, the effects of width $W_2$ of U-shaped microstrip line of top coupling patch on the reflection characteristics of the filter passband have been presented. With the increase of $W_2$, the equivalent quarter-wavelength length of U-shaped SIR increases, which leads to the decrease of low-frequency cutoff frequency, and the two SIR branches whose lengths were $L_2$ outside the U-shaped microstrip line will move outward with the increase of $W_2$. It will affect the coupling with the lower layers and the receiving area of the reflected signal energy from the metal ground plane, so the reflection coefficient and the position of resonance frequency point in the passband will also change with the increase of $W_2$. The in-band reflection coefficient achieves the best situation when $W_2$ = 2.4 mm, which is better than 20 dB.

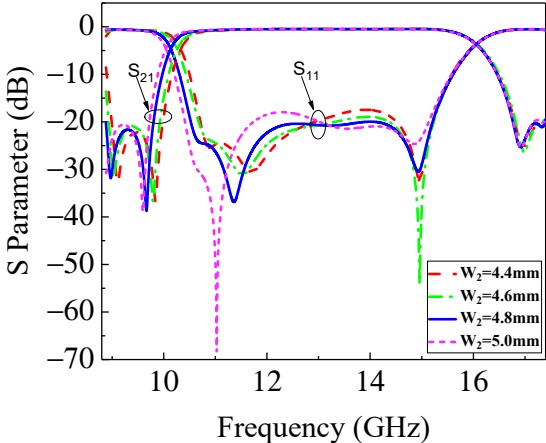

**Figure 6.** Simulation of S parameters when width $W_2$ of the U-shaped microstrip lines is varied from 4.4 mm to 5 mm, with a step of 0.2 mm increment.

Figure 7 shows the influence of diameter $d_0$ of blind holes on high-frequency cutoff frequency and stopband harmonic suppression. Considering limitations of actual processing conditions, the change in $d_0$ should be controlled within an accuracy of 0.1 mm. Generally, the design of blind holes will change coupling strength by adjusting the depth

of the blind holes [26], and there are also designs that add vias or blind holes to change the current path, thus changing the resonance frequency [27]. The multilayer design in this paper needs to add RO4450 prepreg, so the change in depth is limited and the diameter of blind holes is changed for further analysis. As for the SIR on the third substrate, it can be regarded as a quarter-wavelength resonator, and the decrease in its length will lead to an increase in the corresponding high-frequency cut-off frequency. While $d_0$ increases, which is equivalent to the decrease of the quarter-wavelength resonators, the corresponding high-frequency cut-off frequency will move to a higher frequency, and characteristics of high-frequency out-of-band harmonic suppression will also change. As $d_0$ increases, the high-frequency cut-off frequency gradually moves to a higher frequency. Finally, $d_0 = 0.4$ mm is determined as the optimal value under comprehensive consideration.

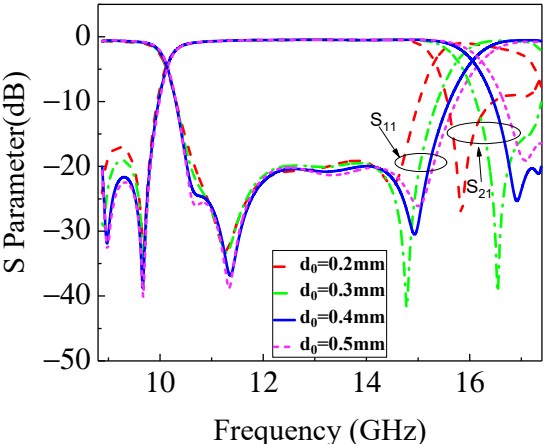

**Figure 7.** Simulation of S parameters when diameter $d_0$ of the blind hole is varied from 0.2 mm to 0.5 mm, with a step of 0.1 mm increment.

As can be seen from Figures 6 and 7, the limit of passband bandwidth tunability mainly comes from the coupling area of multilayer design patterns and the diameter of blind holes. Special attention should be paid to the changes in return loss and stopband suppression caused by their adjustment and adjusting the bandwidth while maintaining the overall performance of the filter.

In Figure 8, effects of width $W_9$ of wider and longer rectangular slots on the resonance points have been presented. Different from those DGS structures composed of multiple identical dumbbell-shaped patterns or some other combination of patterns, the method adopted here is simpler to design and adjust, since the design only consists of three rectangular slots. Two wider and longer rectangular slots of the same size are etched, and then a narrow and short rectangular slot is etched on their symmetry axis. Adjusting the size of three rectangular slots will change the current distribution on the grounded metal surface, which will lead to a change in the equivalent capacitance, and then cause the frequency shift of two resonance points. Of course, the distance between the slots will also have an impact on this. However, the DGS structure corresponds to the SIR on the second floor vertically, and the change in the distance will affect the return loss in the passband, so its parameter adjustment is limited to some extent. Finally, the size of the rectangular slot itself is selected for parameter analysis. In addition, the corresponding resonance frequency will gradually decrease due to the influence of the slow-wave characteristic [28] caused by the blind holes. Therefore, we can find that with the width changing from 1.4 mm to 2 mm, the low-frequency resonance point moves from 11.385 GHz to 11.225 GHz, and the corresponding return loss decreases from −33.44 dB to −38.51 dB. Considering that RL is less than 20 dB, $W_9 = 1.8$ mm is finally selected as the optimal value.

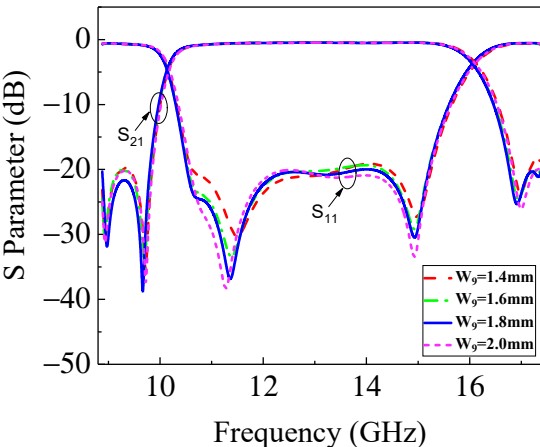

**Figure 8.** Simulation of S parameters when the DGS slot width $W_9$ is varied from 1.4 mm to 2 mm, with a step of 0.2 mm increment.

## 4. Measurement and Simulation Results

The photograph of the manufactured filter is shown in Figure 9a, and its characteristic curve is measured by the vector network analyzer (VNA) 3672D of Ceyear, Qingdao, China. The comparison between measurement and simulation results is shown in Figure 10. The measured 3 dB bandwidth ranges from 10.215–15.375 GHz, RL in the passband is better than 18 dB, roll-off rates are 85.80 dB/GHz and 10.74 dB/GHz respectively, and the insertion loss at the center frequency of 12.795 GHz is 0.58 dB. Measurement results are in good agreement with simulation, and errors in manufacturing and measurement result in some differences in curves.

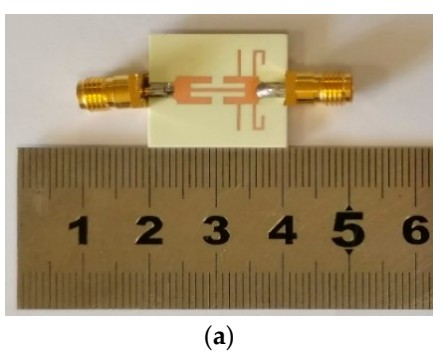

(**a**)

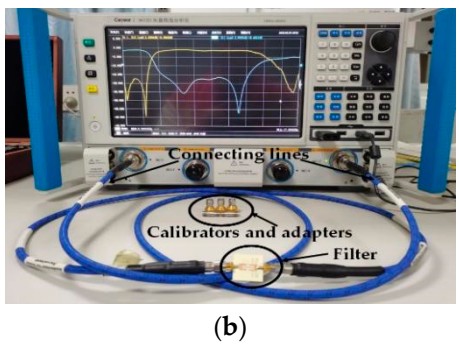

(**b**)

**Figure 9.** Photograph of (**a**) the fabricated UWB filter and (**b**) measurement scene.

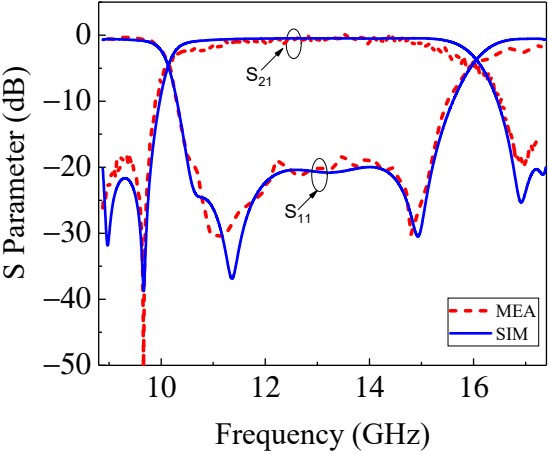

**Figure 10.** Measured S parameters of the proposed UWB filter.

Considering that the length of the quarter-wavelength terminal-open stubs corresponding to low-frequency transmission zero is long, it is folded to reduce the size in the design. The frequency corresponding to the transmission zero is basically consistent with the simulation results, but the stopband suppression characteristics are slightly worse than the simulation results due to the multilayer design and the need to add RO4450F prepreg for processing. After processing, it is necessary to remove the glue residue including spilled resin and other wastes, which can be treated by $CF_4/O_2$ plasma etching and alkaline potassium permanganate.

In addition, the low-frequency resonance point in the passband has a certain frequency shift, which may be due to the fact that when laminating, the prepreg between the two substrates will be squeezed into the holes which leads to an inconsistency between the overall thickness of the device and the thickness of the simulation, and also affects the performance of the blind holes structure. Before measurement, the vector network analyzer must be calibrated with a calibration kit of 31123A (Ceyear, Qingdao, China). During the measurement, as shown in the Figure 9b, the processed filter needs to be connected to SMA connectors and then connected to the vector network analyzer through connecting wires, and the vector network analyzer also needs to be preheated and calibrated. Port 1 of the VNA Ceyear 3672D connects to the input terminal of the filter under test, and port 2 connects to the output terminal of the filter under test, with a frequency sweep range from 8.8 GHz to 17.4 GHz. Errors may also be caused by such preparations not being carried out well. SMA connectors and connecting wires also have a certain insertion loss, which will affect the measurement results.

To reduce the gap between measurement and simulation results, the vector network analyzer needs to be fully warmed up before measurement. The calibration kit is adjusted to the corresponding size of the SMA connector for calibration, and then the observation results are measured several times.

Meanwhile, the proposed filter has a low and flat group delay matched with the tendency of insertion loss, and group delay in the whole passband is less than 0.1 ns, as shown in Figure 11, which ensures the stability and integrity of the signal in the process of transmission.

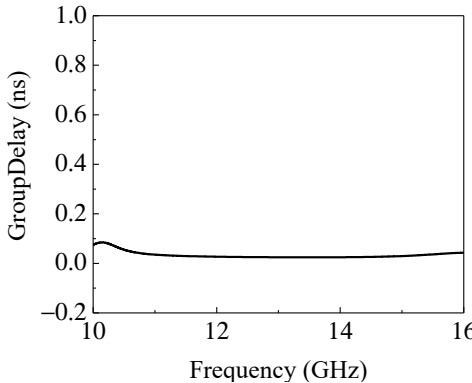

**Figure 11.** Group delay of the proposed UWB filter.

Table 2 shows a comparison between the proposed filter and other designed filters. The UWB filter [2] designed based on traditional methods has a wide bandwidth, but its in-band return loss is large. In Ref. [7], an ultra-wideband bandpass filter is designed by using coupled branch lines, and the odd and even modes of the filter are analyzed by using equivalent circuits. Ultra-wideband and good stopband harmonic suppression are realized. The structure in [13] uses liquid crystal polymer technology to reduce the insertion loss to 0.4 dB, and more transmission zeros are obtained, but the return loss is still large. In Ref. [17], although the structure coated with various oxides achieves a smaller size and lower insertion loss, the bandwidth is relatively narrow. In Ref. [20], a filter with a 3dB FBW of 58% and low insertion loss is realized by a multilayer structure, but its size is too

large, and the design structure is complex. Refs. [21,29] are filters with MPCB structure, which have advantages in performance of stopband, but the insertion loss and return loss in passband are relatively poor. The proposed filter has a compact structure, a relatively small volume, a large fractional bandwidth, narrow roll-off bandwidth, excellent in-band reflection characteristics and competitive insertion loss.

**Table 2.** Comparison of performance with other designed filters.

| Ref. | Technology | 3 dB FBW (%) | $f_0$ (GHz) | $\varepsilon_r$ | IL (dB) | RL (dB) | TZs | Layer | Size (mm$^3$) |
|------|-----------|--------------|-------------|------------------|---------|---------|-----|-------|----------------|
| [2] | HMSIW | 110.1 | 6.85 | 3.55 | <1.6 | >12 | 4 | 1 | $16 \times 14.5 \times 0.508$ |
| [7] | MMR | 62.6 | 4.585 | 3.55 | <2 | >15 | 3 | 1 | $16.88 \times 16.68 \times 0.787$ |
| [13] | LCP | 109.5 | 6.85 | 3.15 | 0.4 | >10 | 7 | 3 | $16.75 \times 19 \times 0.8$ |
| [17] | Hybrid | 10.3 | 3.128 | - | 2.2 | >11 | 2 | 10 | $0.6 \times 0.4 \times 0.506$ |
| [20] | MPCB | 58 | 4.2 | 3/3.66 | 0.95 | >13 | 0 | 3 | $45.05 \times 18.02 \times 2.003$ |
| [21] | MPCB | 35.5 | 5.87 | 3.55 | 1.2 | >15 | 3 | 2 | $50.3 \times 8.6 \times 0.813$ |
| [29] | MPCB | 10 | 10.71 | 2.2 | 1.57 | >15 | 3 | 2 | $23.5 \times 12.6 \times 1.016$ |
| This work | MPCB | 40.33 | 12.795 | 3.66 | 0.58 | >18 | 3 | 3 | $20.8 \times 20 \times 1.34$ |

Compared with other works, it can be found that the cost generally comes from two aspects: materials and processing methods. In comparison with single-layer designs, the multilayer structures have more dielectric substrates and adhesive materials, and the cost will inevitably increase. However, the corresponding multilayer structures also improve their performances, which are mainly reflected in the reduction of insertion loss and return loss in this work. Compared with other LCP or LTCC methods, the cost of MPCB is mainly lower in the processing method. In the works with the same MPCB structure, the Rogers RO4350 dielectric substrate used in this paper is cheaper and the etching pattern is simpler, with only two blind holes, so the overall processing cost is relatively low.

The application scenarios of this work are radio positioning, satellites and radar systems. In these application scenarios, the electronic devices we use are required to integrate a variety of functional elements in a limited space, which requires the integrated electronic devices to have miniaturization characteristics.

All kinds of multilayer structures can reduce the design difficulty and meet the requirements of reducing the size. However, the design based on MPCB will make the location design of vias and material selection of dielectric substrates more extensive, contributing to improving diversification and novelty of design in the future. At the same time, this MPCB method is well-developed now, leading to low cost and easy integration with other contemporary electronic devices, which has great potential in the 5G era.

## 5. Conclusions

A multilayer UWB bandpass filter based on MPCB is proposed in this paper, which has a simple design structure, and is easy to integrate with other electronic devices. The UWB bandpass filter has a 3 dB fractional bandwidth of 40.33%, an IL of 0.58 dB at the center frequency of 12.795 GHz, an in-band RL that is better than 18 dB and a flat group delay of less than 0.1 ns in the passband. The results of measurement and simulation are in good agreement. This filter has a broad application prospect in the fields of radio positioning, satellites and radar systems. In the future, more resonance points or transmission zeros will be introduced to suppress out of band harmonics, achieve better stopband characteristics and improve the utilization of dielectric substrates.

**Author Contributions:** Conceptualization, C.L.; methodology, C.L. and J.-X.C.; software, C.L., J.-X.C., M.-N.W. and J.-M.H.; validation, C.L.; formal analysis, C.L., Z.-H.M. and J.-X.C.; investigation, C.L., M.-N.W. and J.-M.H.; writing—original draft, C.L.; writing—review & editing, C.L. and Z.-H.M.; supervision, Z.-H.M.; project administration, Z.-H.M.; funding acquisition, Z.-H.M. All authors have read and agreed to the published version of the manuscript.

**Funding:** This research was funded by the Fujian Natural Science Foundation Project, grant number 2022J01823.

**Data Availability Statement:** The data presented in this study are available on request from the corresponding author. The data are not publicly available because the data has not been uploaded to publicly archived datasets for the time being.

**Conflicts of Interest:** The authors declare no conflict of interest.

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
