# Peer review of "Design of a Compact Ultra-Wideband Microstrip Bandpass Filter"

_electronics, doi:10.3390/electronics12071728_

Round 1

Reviewer 1 Report

In this paper, authors propose a compact ultra-wideband bandpass filter based on multilayer printed circuit board  (MPCB) structure. In this filter, RO4450F prepreg is used to bond three RO4350B dielectric substrates with different thicknesses. The upper surfaces of the three dielectric substrates are respectively provided with copper-coated structures with different patterns. Two metallized blind holes are designed on which the bottom dielectric substrate based, and a defected ground structure (DGS) consisting of rectangular slots is added to the metal ground layer. Two pairs of quarter-wavelength microstrip terminal-open stubs are loaded in the design of the top coupling patch, which are used to optimize the out-of-band roll-off and stopband suppression, thus introducing two transmission zeros. The DGS provides two resonance points in the passband. The structural parameters of the designed bandpass filter are analyzed, and the simulation and measurement are carried out, which are in good agreement with each other. The insertion loss at the center frequency of 12.795 GHz is 0.58 dB and the fractional bandwidth of 3 dB is 40.33 % from 10.215 GHz to 15.375 GHz. The measurement results show that the proposed filter has good bandpass performance. This bandpass filter can be widely used in Internet of Things, wireless communication and satellite communication.

In my opinion, the paper is interesting, well-written and structured. It is completed of all the needed parts for this kind of topic, a brief description of the proposals in the literature, a detailed description of the proposed structure, measurement results and a comparison with the state of the art.

I want to congratulate the authors for a so interesting paper. 

Author Response

Response: Thank you for your positive encouragement.

Reviewer 2 Report

This paper outlines a straightforward, yet efficient, method to design ultra wideband bandpass filter using multi-layer technology. The method was fairly well-explained and the performance is backed by simulation and measured results. While the paper seems to be helpful for the Microwave community, several aspects must be improved before coming to the final decision on this work. Here are the detailed comments:

The novelty is kind of vague and should be clearly mentioned in the Abstract and the last paragraph of the introduction.

Please explain the limit of the bandwidth tunability

Measured and simulated results [Fig 7]: There are some discrepancies in this fig between the measurement and simulations. While they are not fully justified. Additionally, please explain how those errors can be minimized in the design and fabrication process. 

The introduction does not cover the state-of-the-art in relation to the various design approaches of bandpass filters. In detail, one of the most important and well-known approaches in BPF design is even and odd mode analysis as explained in: triple mode spiral wideband bandpass filter using symmetric dual-line coupling, Electronics Letters. Another missing approach is using the transfer function of the filter to predict the frequency behaviour of the filter as explained in: design of a compact planar transmission line for miniaturized rat-race coupler with harmonics suppression. Another example can be found in : analysis of novel approach to design of ultra‐wide stopband microstrip low‐pass filter using modified u‐shaped resonator.

Please give more detail regarding the measurement process and settings.

Add more results: Please add current distribution at some critical frequencies and comment the results accordingly

Some very recent methods are also neglected in the intro that should be covered, for instance, a very recent method for designing highly controllable filters are based on novel metamaterials as explained in: Tunable terahertz filter/antenna-sensor using graphene-based metamaterials. In fact graphen-based circuits have shown very good performance for a varity of microwave applications, including electromagnetics filtering as explained in: Recent and emerging applications of Graphene-based metamaterials in electromagnetics.

Please explain how the stopband of the filter can be improved

No comments in the paper about fabrication cost. Please elaborate

The conclusion is way too long and needs to be revised and shortened. Lots of unnecessary information is there. 

The English of the paper needs improvement. There are not such serious grammar mistakes, but many sentences could be written in a better way.

Please compare your design with the latest works here is one recent work that can be used

"Varactor-tuned wideband band-pass filter for 5G NR frequency bands n77, n79 and 5G Wi-Fi", Scientific Reports 12 (1), 1-10. 

The equivalent circuit model is one of the greatest ways to provide a good insight into the filtering mechanism. The authors don’t have to necessarily provide a LC model, instead they can refer to literature. Here is one recommended reference for this purpose

"All-metal wideband frequency-selective surface bandpass filter for TE and TM polarizations". Transactions on Antenna and Propagation, 2022, DOI: 10.1109/TAP.2021.3138256 

Reviewer 3 Report

The authors have designed a UWB BP filter, which is a outdated one.

Novelty of the design is very low. Only materials changed in the design.

This article shall be added with contents of Application based filter design. 

Bandwidth is increased from 10.215 GHz to 15.375 GHz. Justify using Mathematical model. 

Reviewer 4 Report

The paper discusses an interesting topic and the technical work sounds good. Some comments to improve the paper;

1. In regards to your finding, you mentioned that the proposed work provides a great advantage in term of size, please also discusses on how this size will provide a great advantage in the specific application as the smaller size does not really a matter for some applications.

2. When you said the design is relatively simple, does it means in term of process or the approach used. Please clarify on this matter.

3. Please recommend some future works in your conclusion.

Round 2

Reviewer 2 Report

I dont see any further concerns in this paper. It can be accepted now.

Reviewer 3 Report

NA